# The Development of an Assembled Truss Core Lightweight Panel and Its Method of Manufacture

Zhilei Tian [1], Chenghai Kong [2], Jingchao Guan [1], Wei Zhao [3], Apollo B. Fukuchi [1] and Xilu Zhao [1,*]

1   College of Mechanical Engineering, Saitama Institute of Technology, Saitama 369-0293, Japan
2   Topy Industries Co., Ltd., Aichi 441-8510, Japan
3   Weichai Global Axis Technology Co., Ltd., Tokyo 107-0062, Japan
*   Correspondence: zhaoxilu@sit.ac.jp

**Abstract:** In this study, a new assembled truss core panel and the method for processing it were proposed in order to improve the performance of the lightweight panel structure. The proposed assembled truss core panel can be easily processed by simple punching and bending. A processing experiment on an assembled truss core panel was conducted using an aluminum plate with a thickness of 1.0 mm, and the validity and performance of the proposed processing method were verified. A three-point bending test was performed using an assembled truss core panel obtained using the processing experiment. The assembled truss core panel had a relatively high bending stiffness in its early elastic deformation and a relatively long-lasting bending deformation after the initial failure. Its application as a lightweight panel has been confirmed. In order to compare it with the most commonly used honeycomb lightweight panel, FEM (finite element method) analysis was performed on the assembled truss core panel and on the honeycomb panel under the same conditions. The bending stiffness of the assembled truss core panel was found to be 10.60% higher than that of the honeycomb panel. Furthermore, to improve the productivity of the assembly-type truss core panel, construction of a production line using progressive dies was proposed, and the possibility of practical development for mass production was examined.

**Keywords:** truss score panel; assembled manufacturing method; origami engineering; lightweight panel structure; bending forming; riveted structure; mechanical properties

## 1. Introduction

Research and development of high-performance lightweight structures is an important goal in the industrial world, and various new lightweight structures have been proposed and studied [1–7]. The use of composite materials and aluminum alloys instead of metallic materials to realize lightweight structures is being studied. However, factors such as high manufacturing costs have prevented such structures from being put to practical use [8–10].

Honeycomb structures have conventionally been used as lightweight structures in industries such as transportation and machinery [11,12]. The static and dynamic mechanical properties of honeycomb panels have been investigated using FEM (finite element method) analysis and by performing measurement experiments [13–15]. Strength issues in honeycomb panel structures with local defects caused by stress concentration have also been investigated [16,17]. The compressive buckling strength of honeycomb panels, which is important for thin-walled structures, has also been investigated in respect of the problem of crushing [18]. Furthermore, along with flat honeycomb panels, honeycomb structures with cylindrical shapes and arbitrary cross-sectional configurations suitable for special applications have also been proposed [19,20]. These studies provide a valuable technical basis for applying honeycomb panels to high-strength structures in the industry.

The honeycomb panel has a core material with a regular hexagonal cross-section and face plate, which are bonded together with glue. However, its disadvantage is that it cannot

be applied to structures subject to large-scale vibrations and shear loads. To rectify this problem, a truss core panel composed of triangular trapezoidal cells was proposed [21,22]. Truss core panels have been studied through simulations and have been found to be mechanically superior lightweight structures [23–25].

High-quality truss core panels cannot be manufactured at a low cost owing to their complicated three-dimensional shape. The use of a thin-steel stamping method to form the truss core panels might be feasible. To increase the initial bending stiffness, the truss core part must be formed as high as possible by deep drawing. However, if the truss core part is very high, the wall becomes extremely thin locally. A progressive press-forming method using a hemispherical or hexagonal frustum intermediate model has been proposed, and this has alleviated the problem of the obtained truss core panel being extremely thin locally [26–29].

However, when forming a truss core from a thin steel plate, a stable press-forming method has not been developed, because the problem of the core becoming extremely thin locally cannot be avoided.

On the other hand, processing methods such as sandwich core materials using truss beams and bent plates have been proposed as an alternative to press molding [30–32]. Among them, a lightweight structure based on a space beam structure made of non-metal material such as aluminum alloy is being studied [33]. However, in the case of a lightweight panel structure in which high bending stiffness is pursued, as in this study, a core material using a bent plate is superior to a truss beam. There are, however, no examples of truss core panels using bent plates that have been developed. It is necessary to consider a new truss core panel processing method.

In this study, a new processing method using origami engineering for prefabricated truss core panels is proposed as an alternative to the press-forming method. Firstly, a flat plate part obtained by opening the truss core shape is prepared. Next, the flat plate parts are bent and, with the use of glue and rivets, are formed into truss cores. Finally, the truss core and the face plate are joined with glue and rivets to obtain a truss core panel. The truss core panel processed by the proposed method does not reduce the plate thickness, and since it is connected between the truss cores with glue and rivets on the inclined surface, it can also improve the initial bending stiffness of the panel structure. In this study, we conducted a fabrication experiment on the proposed assembled truss core panel and examined its processing performance. The mechanical performance of the fabricated assembled truss core panel was examined in detail using FEM analysis and the three-point bending test.

## 2. Materials and Methods

### 2.1. Assembled Truss Core Panel

The truss core panel shown in Figure 1 is constructed by the orderly arrangement of the truss core parts on a flat plate. A truss core has a hexagonal truncated pyramidal shape with three trapezoidal sides and three rectangular sides and is represented by these shape parameters: height $h$, base length $a$, and side length $b$ [29]. A double truss core panel is constructed by connecting two truss core panels with opposite sides facing each other, as shown in Figure 2. When a load is applied from the out-of-plane direction, the load tends to spread to the truss core panel through the slanted contact surfaces of the truss cores. The truss core panel has good initial bending stiffness and vibration characteristics.

However, to increase the rigidity of the truss core panel, the truss core must be formed high; this is not easily achieved.

To solve this problem, a two-step deep drawing method using a progressive die was proposed [29]. In the first step, the flat plate material shown in Figure 3a was formed into the shape of the intermediate model shown in Figure 3b. In the second step, the final truss core panel was molded from the shape of the intermediate model. Thus, the truss core panel with a relatively high height $h$ was formed using a two-stage deep drawing process, as shown in Figure 3c.

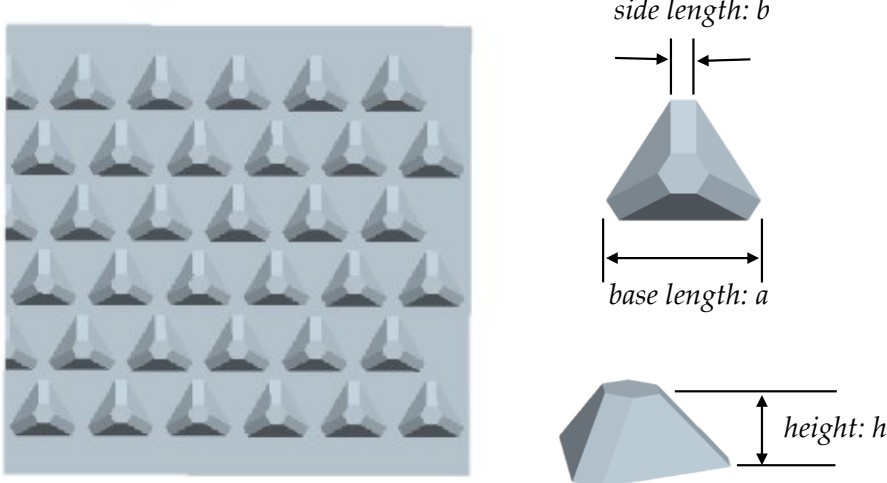

**Figure 1.** Geometry of the truss core panel and truss core shape parameters.

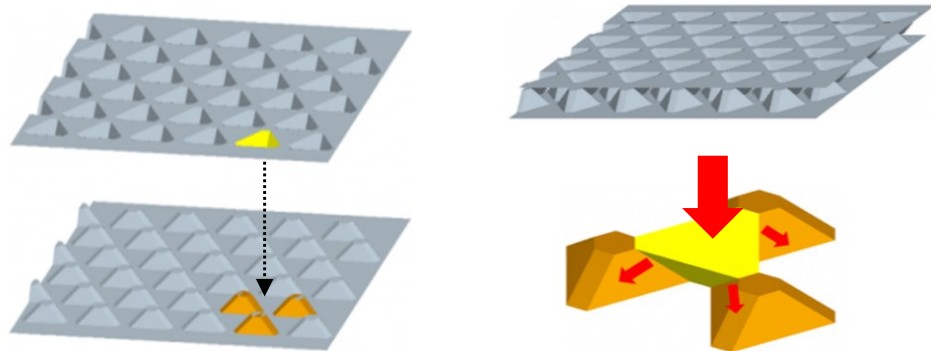

**Figure 2.** The double truss core panel can diffuse the load through the diagonal surface of the truss core.

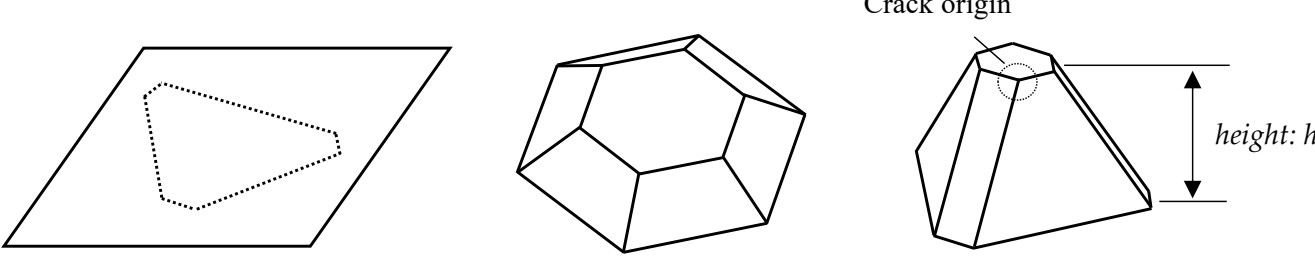

**Figure 3.** A truss core panel formed using a two-stage progressive deep drawing method.

However, the area of the planar portion before forming, indicated by the dotted line in Figure 3a, was increased to the surface area of the three-dimensional truss core in Figure 3c after forming, and the thickness of the truss core was reduced. In particular, the area around the bottom apex of the truss core, indicated by the dotted line in Figure 3c, was the thinnest, and it was there where cracks could occur. Localized thinning of the truss core could also adversely affect the mechanical properties of the truss core panel.

Therefore, processing the truss core panel using the deep drawing method is problematic, and no examples of its practical usage exist.

To overcome the shortcomings of the deep drawing method in processing truss core panels, we propose a new assembled truss core panel, as shown in Figure 4.

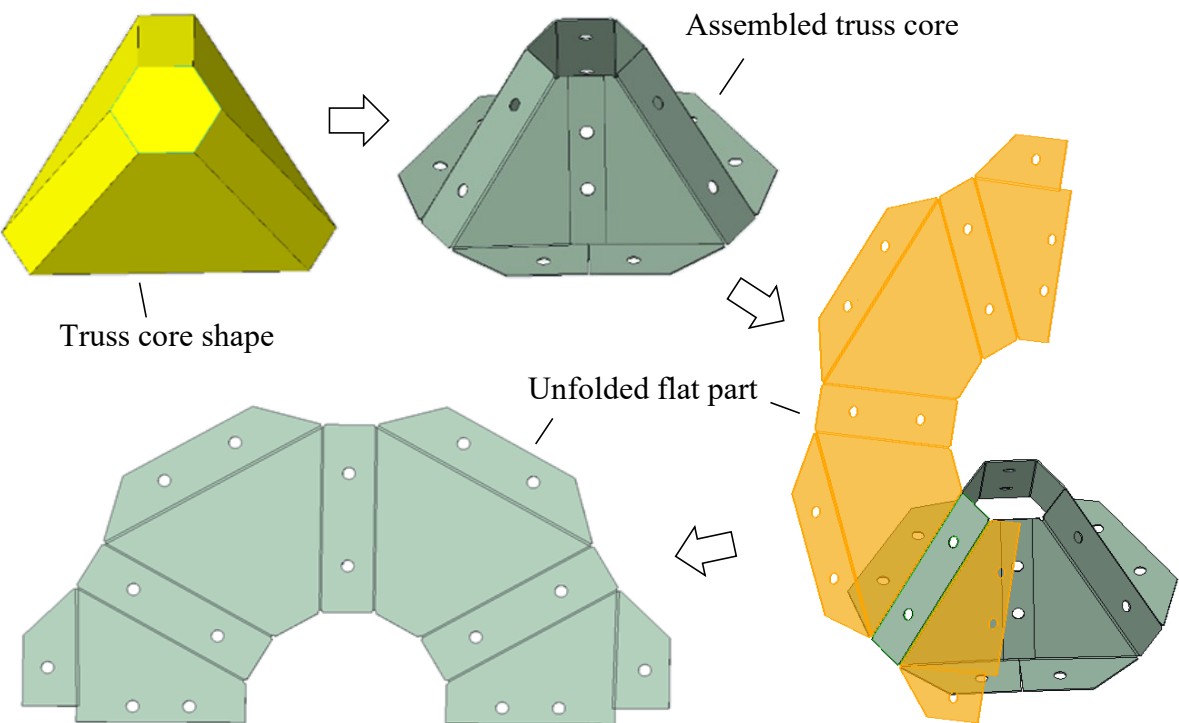

**Figure 4.** Construction of the flat plate parts of the assembled truss core.

The process of constructing the planar parts of the assembled truss core is shown in Figure 4. Firstly, an assembled truss core was created from the truss core geometry. The planar part was then constructed by opening the assembled truss core geometry.

Assembled truss core panels were formed as shown in Figure 5. Firstly, the planar part of the truss core was bent. The joints were glued and riveted to obtain a single truss core. Subsequently, the four single truss cores were alternately combined. A truss core unit was obtained by attaching the oblique joints between the truss cores with glue and rivets. Thereafter, several truss core units were joined with glue and rivets to obtain a component of the assembled truss core. The final assembled truss core panel was thus obtained by gluing and riveting face plates to both sides of the assembled truss core members.

In contrast to the conventional deep drawing method, the production of assembled truss core panels using the procedure shown in Figure 5 does not reduce the thickness of the material, and it solves the problem of cracks occurring during the manufacturing process. It also improves the mechanical properties of assembled truss core panels joined using glue and rivets.

### 2.2. Design parameters

In order to realize the processing method of the proposed assembly-type truss core panel, the first task was to design the planar part from the shape parameters of the truss core, as shown in Figure 1.

The truss core shape can be designed according to the 3 steps shown in Figure 6. In step 1, the plane was divided into equilateral triangles of side length *a*, as shown in Figure 6a. In step 2, each of the three corners was cut with width *b* to obtain equilateral triangular and hexagonal shapes, as shown in Figure 6b. In step 3, the equilateral triangle was translated vertically to a height *h* to obtain a three-dimensional truss core shape. The truss core consists of a regular hexagonal top face, three rectangular sides, and three trapezoidal sides. The geometric parameters of the truss core are *a*, *b*, and *h*.

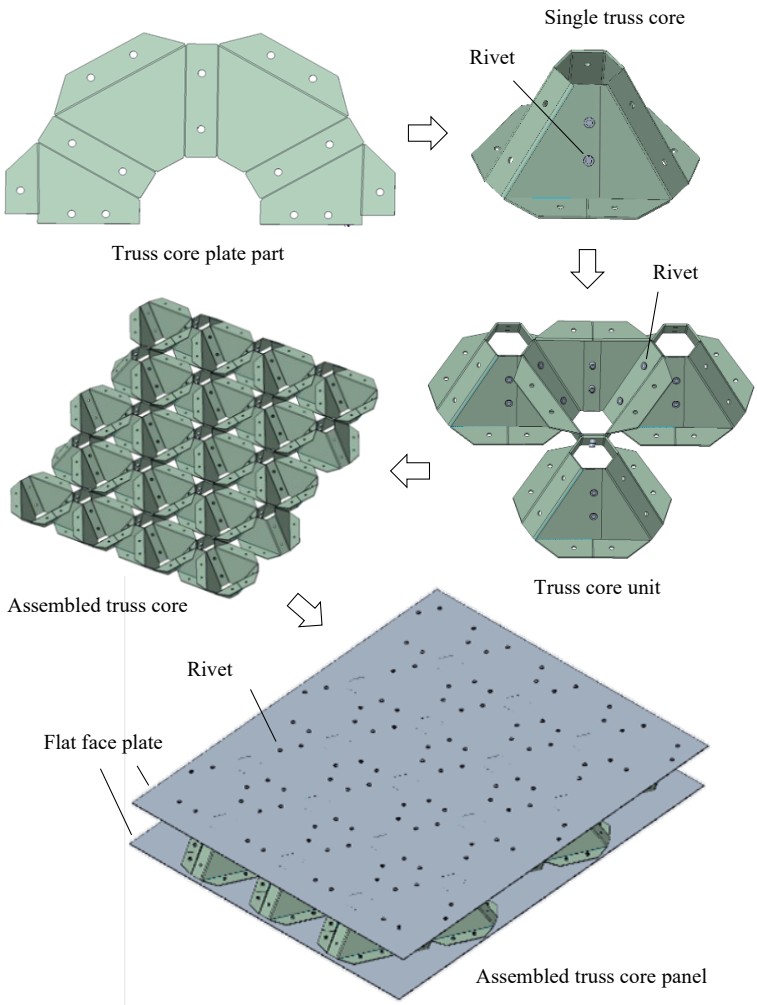

**Figure 5.** Creation process of assembled truss core panel structure.

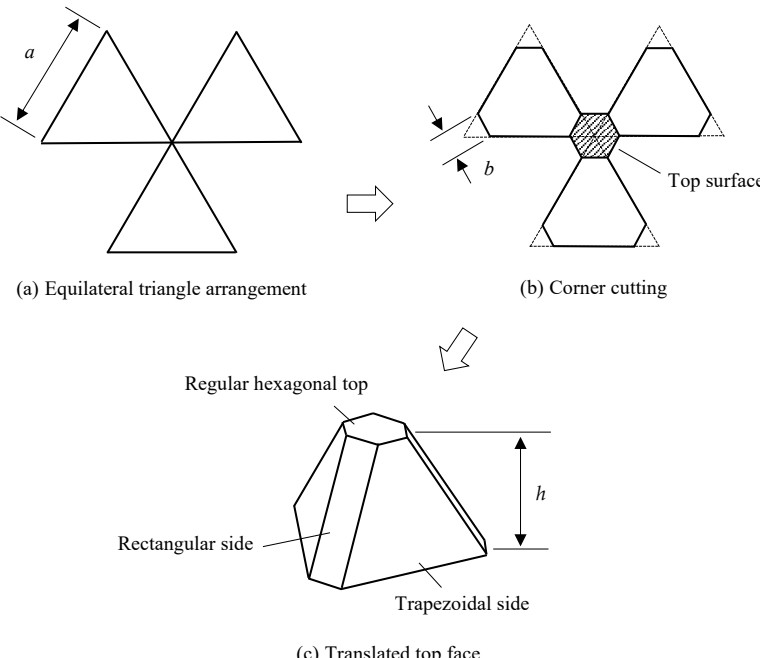

**Figure 6.** Design process of the truss core shape. The basic shape parameters are *a*, *b*, and *h*.

Figure 7 shows an enlarged view of the truss core designed in Figure 6. From the right triangle $\Delta oqr$ on the top surface, the side length $oq$ is obtained using the following equation:

$$oq = b \sin 60° = \frac{\sqrt{3}}{2}b, \tag{1}$$

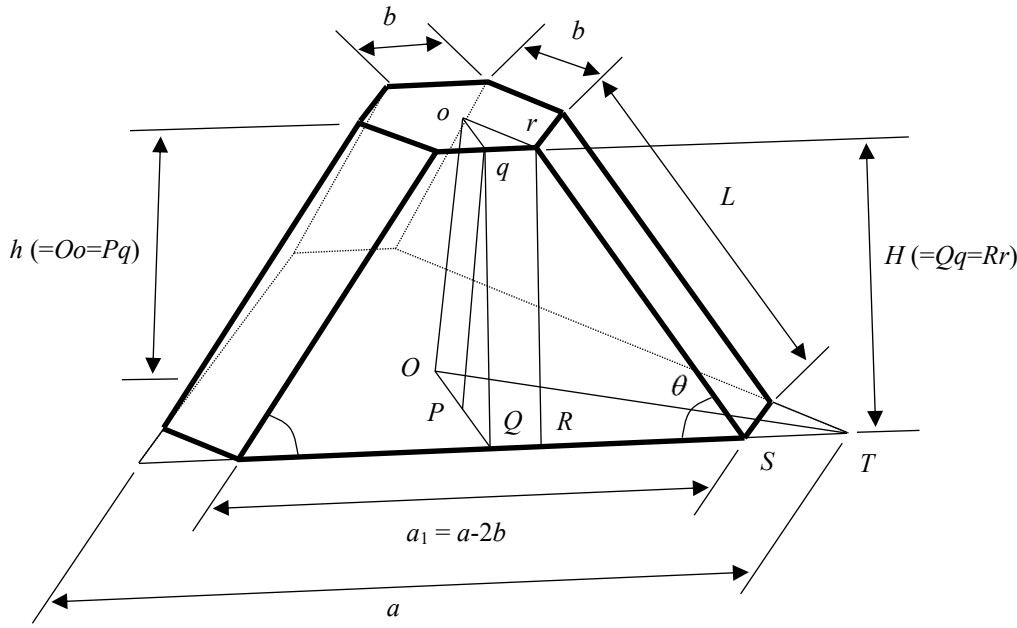

**Figure 7.** Diagram of the shape parameters of the flat part with truss core unfolded.

With the right triangle $\Delta OQT$ on the base, the side length $OQ$ is obtained using the following equation:

$$OQ = \frac{a}{2} \tan 30° = \frac{\sqrt{3}}{6}a, \tag{2}$$

Using Equations (1) and (2), the height $H$ of the side trapezoid is obtained from the right triangle $\Delta qPQ$ using the following equation:

$$H = \sqrt{(OQ - oq)^2 + h^2} = \sqrt{\frac{(a - 3b)^2}{12} + h^2}, \tag{3}$$

From the right triangle $\Delta rRS$, the side length $L$ and the angle $\theta$ of the side trapezoid are obtained using the following equation:

$$L = \sqrt{\left(\frac{a - 3b}{2}\right)^2 + H^2} = \sqrt{\frac{(a - 3b)^2}{3} + h^2}, \tag{4}$$

$$\theta = \sin^{-1}\frac{H}{L} = \sin^{-1}\left(\frac{1}{2}\sqrt{\frac{(a - 3b)^2 + 12h^2}{(a - 3b)^2 + 3h^2}}\right), \tag{5}$$

In addition, the length $a_1$ of the base of the side trapezoid is obtained using the following equation:

$$a_1 = a - 2b, \tag{6}$$

The shape parameters $a_1$, $b$, $H$, $L$, and $\theta$ of the plane part shown in Figure 7 can be calculated by directly substituting the shape parameters $a$, $b$, and $h$ of the truss core shown in Figure 6 into Equations (3)–(6).

Furthermore, based on the basic shape of the planar part shown in Figure 8, a partial shape and joining holes for rivets can be added to create a blueprint for the truss core flat plate part, as shown in Figure 9.

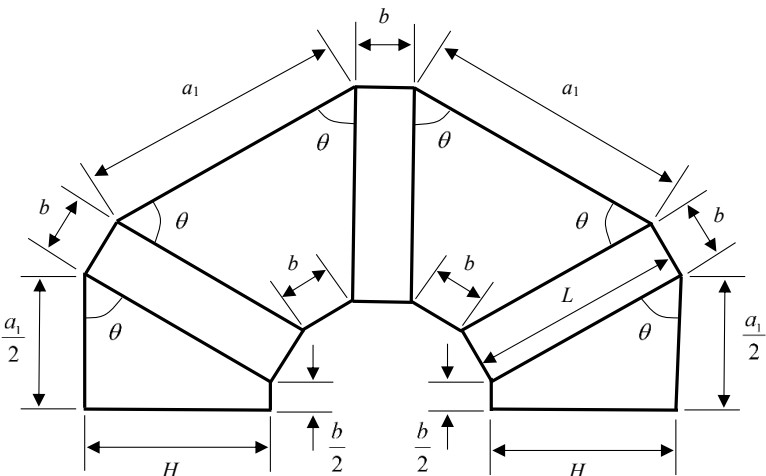

**Figure 8.** Basic shape and size of the truss core plane part. The flat part parameters are $a_1$, $b$, $H$, $L$, and $\theta$.

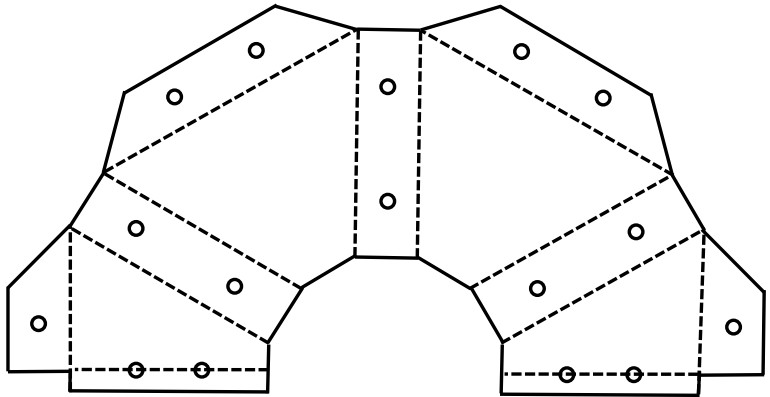

**Figure 9.** Schematic of the planar part of the truss core.

### 2.3. Manufacturing Test

The assembled truss core panel was processed using the proposed method shown in Figure 5. Table 1 shows the details of the truss core panel used in the processing experiments in this study.

**Table 1.** Details of the truss core panel examined in this study.

| | |
|---|---|
| Base triangle side length $a$ | 70 mm |
| Corner cut side length $b$ | 10 mm |
| Truss core height $h$ | 33 mm |
| Plate thickness $t$ | 1.0 mm |
| Material of face plate and truss core | Aluminum A5052 |
| Truss core panel size | 350 mm × 300 mm |

Figure 10 shows the state of the processing experiment on the assembled truss core panel. The plane parts were cut with a laser processing machine according to the CAD shape

data. A bending tool was used to bend the planar part into the shape of a truss core along the fold lines, and the folded parts were attached with glue and rivets to obtain a single truss core. Adjacent truss cores were affixed with glue and rivets along the rectangular sides of the truss cores to obtain a double truss core set. Similarly, several adjacent truss cores were riveted together and secured with an adhesive to form an assembled truss core member. Finally, flat plates were glued and riveted to each side of the assembled truss core member to obtain the final assembled truss core panel.

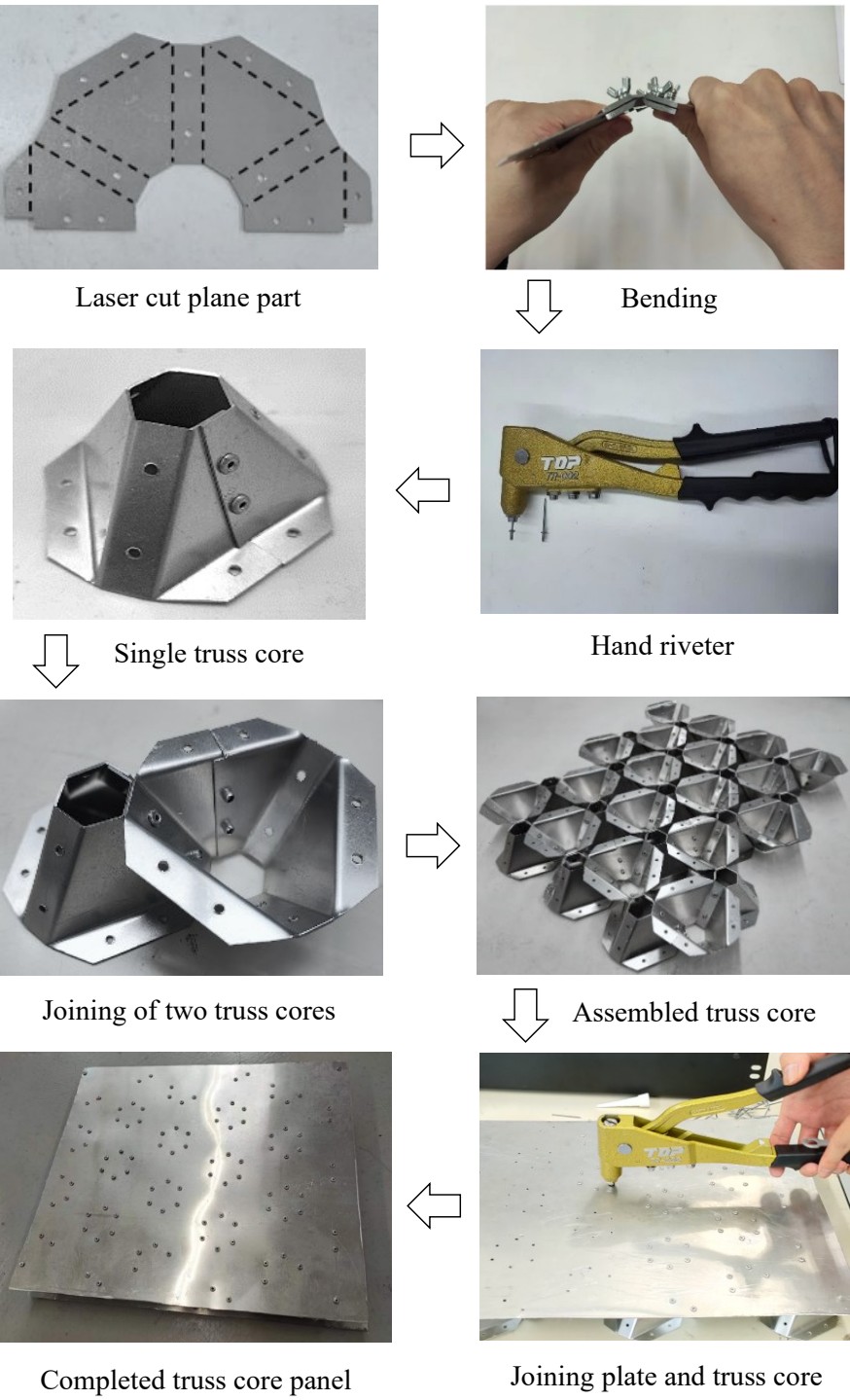

**Figure 10.** Processing experiment of an assembled truss core panel.

## 3. Results and Discussions

### 3.1. Experimental Verification of Mechanical Properties

A three-point bending test was used to study the mechanical properties of the assembled truss core panel, as shown in Figure 11. The autograph machine used was a Shimadzu AG-300kNG. The side length of the assembled truss core panel was 350 mm × 300 mm and its thickness was 34 mm. The support width for the three-point bending test was 200 mm. A square steel pipe with a side length of 50 mm was used for the three-point bending jig, which was in contact with the assembled truss core panel. The corners were rounded to 10 mm.

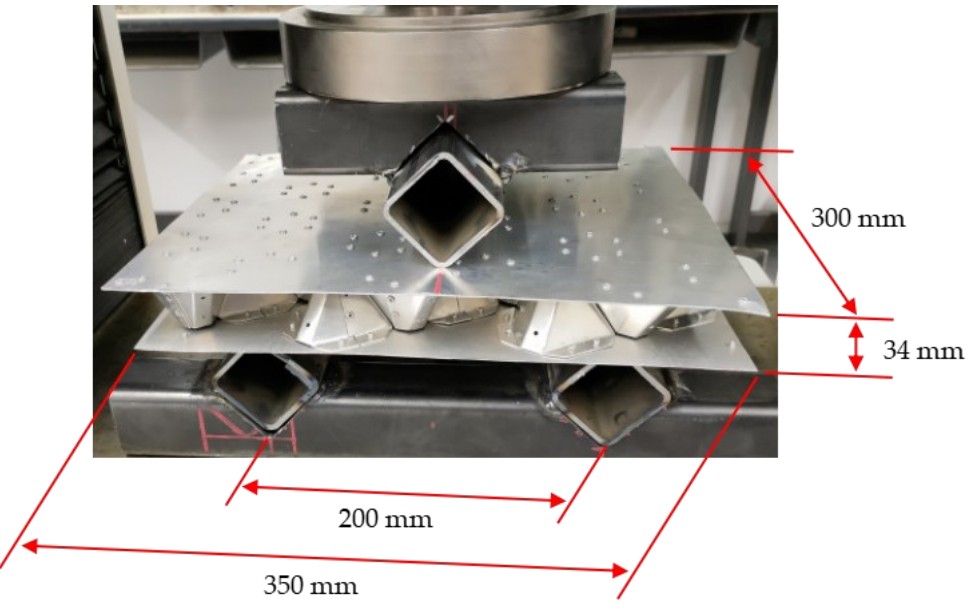

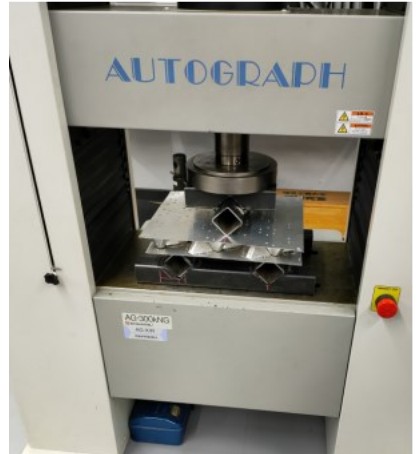
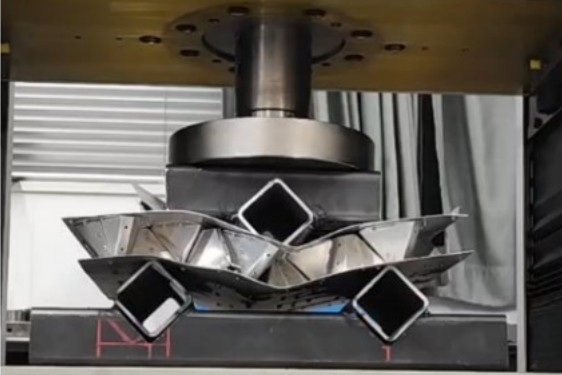

**Figure 11.** Three-point bending test to examine mechanical properties of assembled truss core panel.

The application speed of the bending load was 10 mm/min, and the maximum displacement of the load was 20 mm. The recording frequency of the measurement data was 1000/s.

In comparing this method with the conventional deep drawing method, the major problem affecting the assembled truss core panel was its joint strength. As shown in Figure 12, in the three-point bending test, joint rivets broke frequently. Using glue and rivets together was effective in improving the rate of joint rivet breakage.

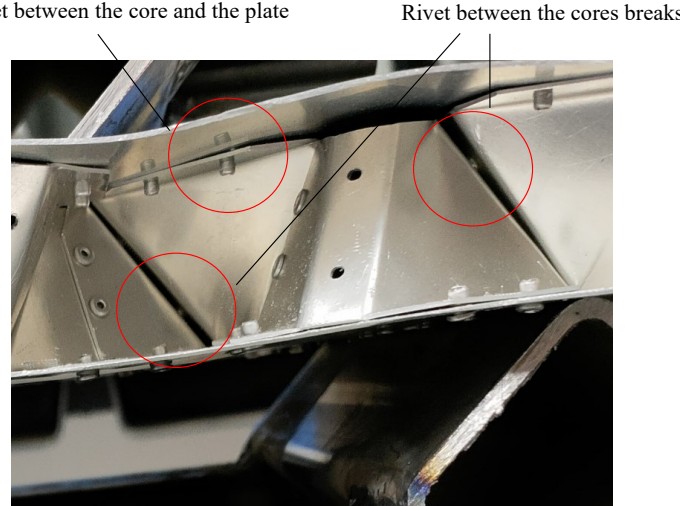

**Figure 12.** Partial enlargement of three-point bending test.

To test this strategy, two different truss core panels were prepared under the same conditions, with one truss core panel joined solely with rivets and the other truss core panel joined using both glue and rivets, as shown in Figure 13. The glue used here is the AX-041 chemically reactive glue manufactured by the CEMEDINE company.

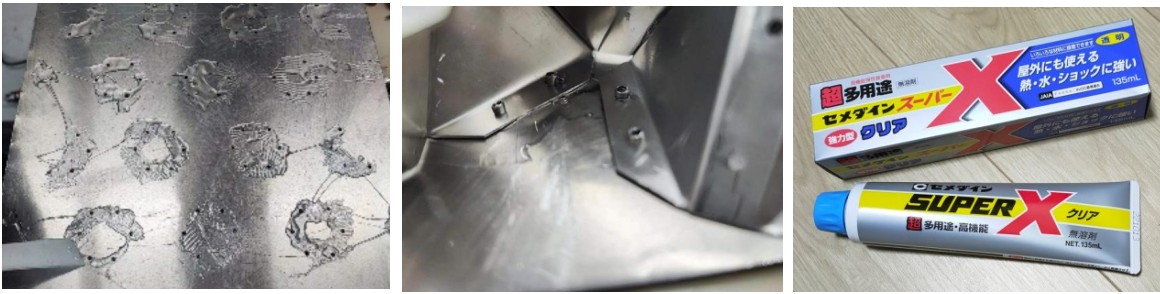

**Figure 13.** Assembled truss core panel joined using glue and rivets.

A similar three-point bending test was performed to examine the mechanical properties of each panel.

Figure 14 shows the results of a three-point bending test conducted on the two assembled truss core panels joined using different methods. The blue and red lines represent the results for the assembled truss core panel joined with rivets alone and the assembled truss core panel joined using both glue and rivets, respectively.

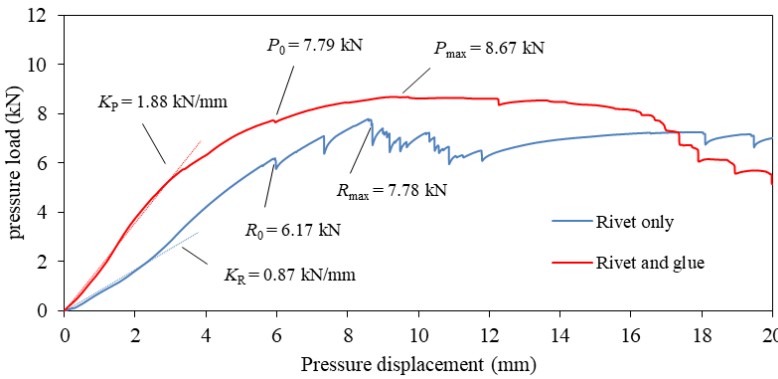

**Figure 14.** Three-point bending test results of assembled truss core panels joined by different methods.

As indicated by the dotted line in Figure 14, the initial bending stiffness of a lightweight panel is important. The initial bending stiffness of the truss core panel joined with glue and rivets at $K_P$ = 1.88 kN/mm is 161.1% higher than that of the truss core panel joined only with rivets at $K_R$ = 0.87 kN/mm. Therefore, in the case of glued and riveted truss core panels, the load is spread over the entire oblique surface of the adjacent truss core. For truss core panels joined only with rivets, the load is spread only over the adjacent rivets. Different load transmissions between adjacent truss cores account for the difference in the initial bending stiffness.

As shown in Figure 14, during deformation, the load drops suddenly at several points, these being the points where rivets fell out of the joined plates.

Comparing the two graphs in Figure 14, it is evident that the rivets first fell off at a displacement of approximately 6 mm. This is related to the fact that both truss core panels have the same internal geometry. However, as the adhesive has a bonding action, when both the adhesive and rivets are used for bonding, the number of occurrences of the rivets falling off is relatively low, as is the amount of load drop.

The load when the joining rivet first came off was $P_0$ = 7.79 kN when glue and rivets were used, whereas it was $R_0$ = 6.17 kN when only rivets were used. Thus, bonding with glue and rivets allows a 26.3% higher load than bonding with rivets alone.

The maximum load of the assembled truss core panels joined using both glue and rivets was $P_{max}$ = 8.67 kN, which was 11.4% higher than the maximum load $R_{max}$ = 7.78 kN when only rivets were used.

Therefore, the assembled truss core panel joined by glue and rivets has a relatively high initial bending stiffness in the initial elastic deformation, and the bending deformation continues in the latter period for a relatively long time after the initial failure, indicating that it can be applied as a general-strength panel structure.

For confirmation, an FEM model was created according to the assembled truss core panel construction shown in Figure 10, and an analysis was performed by applying a three-point bending load as shown in Figure 11. Table 2 shows the comparison results. The initial bending stiffness of the truss core panel using rivets and glue was 2.01 kN/mm, and the initial bending stiffness of the truss core panel using only rivets was 0.93 kN/mm. Comparing these analysis results with the experimental results in Figure 14, it was observed that the analysis value for the truss core panel using rivets and glue was 6.91% higher, and the analysis value for the truss core panel using only rivets was 9.20% higher. The reason for this is that the bond condition of the FEM analysis model seems to be stronger than that of the actual structure.

**Table 2.** Comparison of initial bending stiffness results for assembled truss core panel.

|  | Analysis (kN/mm) | Experiment (kN/mm) | Change |
|---|---|---|---|
| rivets and glue | 2.01 | 1.88 | 6.91 % |
| only rivets | 0.93 | 0.87 | 9.20 % |

*3.2. Comparison with Conventional Lightweight Panel*

Assuming the elastic conditions applied to lightweight panel structures, the bending stiffness of the assembled truss core and honeycomb panel were compared using the FEM analysis results shown in Figure 15.

As shown in Figure 15, a uniformly distributed load, $q$ = 1000 N/m$^2$, was applied to a square panel with a side length of 900 mm and a thickness of 30 mm, with all boundaries fixed. The core size of these panel structures was also 30 mm. The software Ls-dyna was used for the FEM analysis [34]. The average edge length of the analysis mesh was 4 mm. The rivet model used beam elements of the same cross-sectional area, and the bond model was used for the glue joint. Detailed parameters of the FEM analysis model are shown in Table 3.

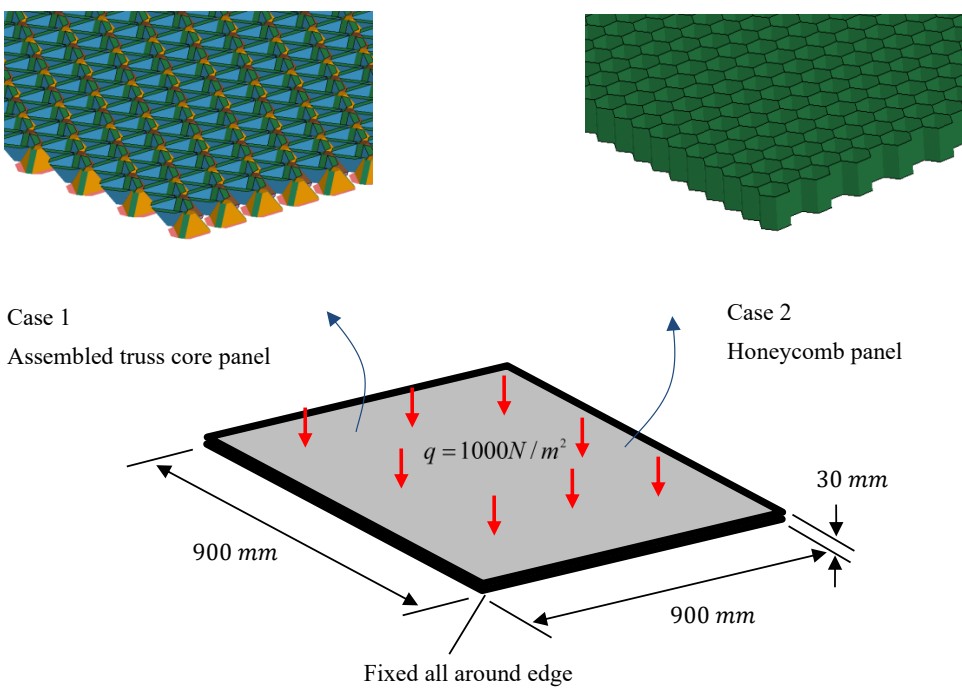

**Figure 15.** FEM analysis model of assembled truss core panel and honeycomb panel.

**Table 3.** Detailed parameters of FEM analysis model.

| Item | | Parameter |
|---|---|---|
| Material | Aluminum | A5052 |
| | Young's modulus | 70 GPa |
| | Poisson's ratio | 0.33 |
| | Thickness | 0.8 mm |
| Assembled truss core panel | Nodes | 142342 |
| | Elements | 154385 |
| Honeycomb core panel | Nodes | 124223 |
| | Elements | 100241 |

Under the same numerical analysis conditions, the analysis results of the assembled truss core panel and honeycomb panel are shown in Figure 16 and Table 4, respectively.

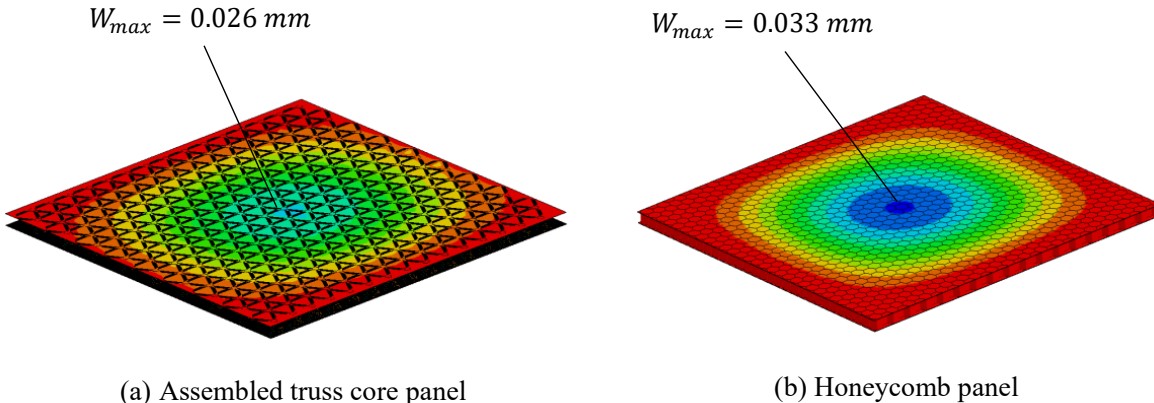

$W_{max} = 0.026\ mm$

$W_{max} = 0.033\ mm$

(a) Assembled truss core panel

(b) Honeycomb panel

**Figure 16.** Analysis results of two types of lightweight panels under the same conditions.

**Table 4.** Comparison of analysis results for assembled truss and honeycomb core panel.

|  | Assembled Truss | Honeycomb | Change |
|---|---|---|---|
| Maximum deflection at center point | 0.026 mm | 0.033 mm | −21.2% |
| Initial bending stiffness | 31,153.85 kN/mm | 24,545.45 kN/mm | 26.9% |
| Initial bending stiffness per unit mass | 3084.54 kN/(kgmm) | 2789.26 kN/(kgmm) | 10.6% |

The maximum deflection of the assembled truss core panel was 0.026 mm and that of the honeycomb panel was 0.033 mm. The maximum deflection of the assembled truss core panels was 21.2% less than that of the honeycomb panels.

The initial bending stiffness can be evaluated by the load value per unit maximum deflection. The initial bending stiffness values of the assembled truss core panel and honeycomb panel were 31,153.85 KN/mm and 24,545.45 KN/mm, respectively. The initial bending stiffness of the assembled truss core panel was 26.9% greater than that of the honeycomb panel.

However, since the weights of the two types of lightweight panels are different, it is more informative to compare the initial bending stiffness values per unit mass. The initial bending stiffness per unit mass of the assembled truss core panel was 3084.54 KN/(kgmm) and that of the honeycomb panel was 2789.26 KN/(kgmm). Therefore, the initial bending stiffness per unit mass of the assembled truss core panel was 10.60% higher than that of the honeycomb panel.

The two reasons for the initial bending stiffness of the assembled truss core panel being higher than that of the honeycomb panel are as follows. (1) The riveted wing material at the bottom of the truss core shown in part *A* of Figure 17 was directly integrated with the face plate. The wall of the face plate became relatively thick, resulting in an increase in the moment of inertia of area of the assembled truss core panel. (2) As shown in part *B* of Figure 17, the load was distributed over a wide area of the truss core panel through the oblique rectangular sides between the truss cores. In respect of this difference, and as shown in Figure 16, the bending deformation was relatively concentrated in the center of the honeycomb panel.

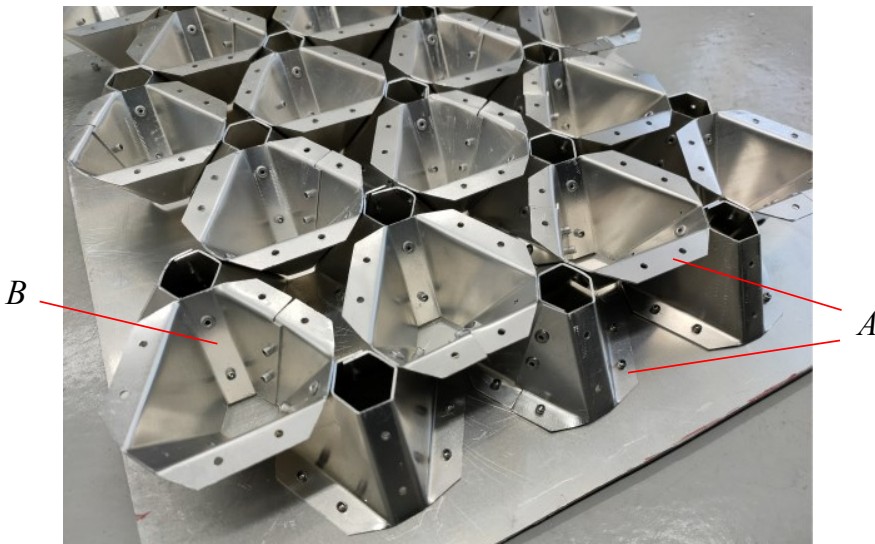

**Figure 17.** Joints between truss cores and joints between truss core and face plate.

### 3.3. Comparison with Conventional Forming Method

Let us consider a truss core panel, as shown in Figure 6, formed using the conventional deep drawing method. The hexagonal area indicated by the dotted line of the flat plate before forming, as shown in Figure 18a, can be calculated using the following equation:

$$A_{before} = \frac{\sqrt{3}}{4}a^2 - \frac{3\sqrt{3}}{4}b^2, \tag{7}$$

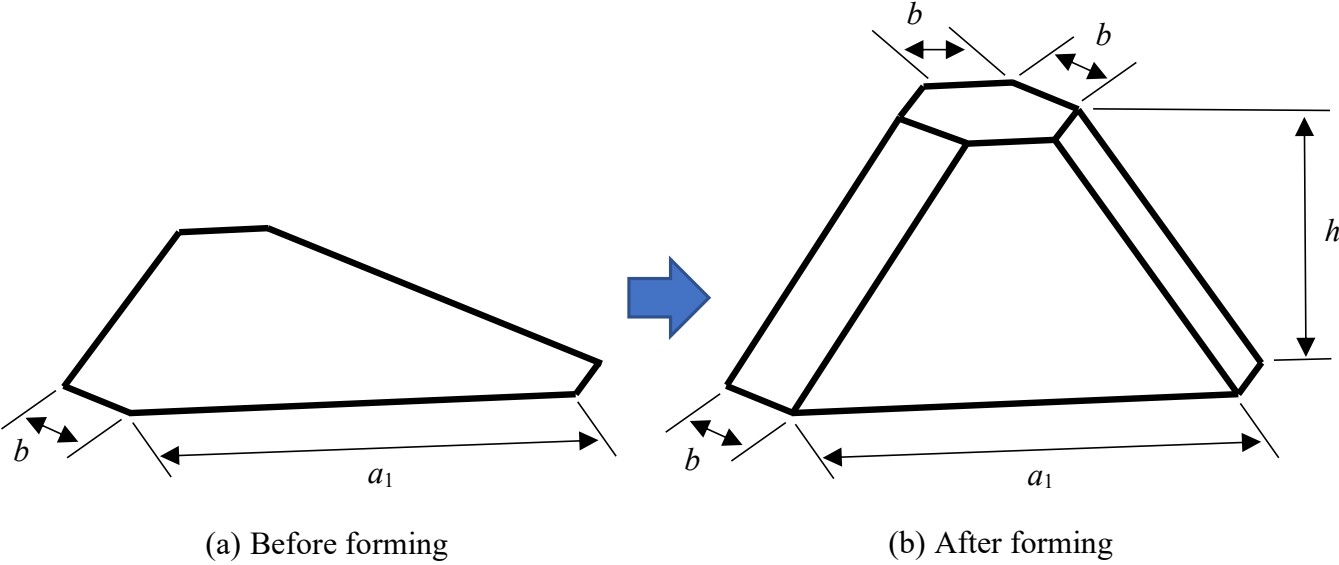

(a) Before forming                    (b) After forming

**Figure 18.** Surface area change when forming a truss core using the deep drawing method.

The formed three-dimensional truss core is shown in Figure 18b, and its surface area can be calculated using the following equation:

$$A_{after} = \frac{3}{2}(a_1 + b)H + 3Hb + \frac{3\sqrt{3}}{2}b^2, \tag{8}$$

Given that the material volume before and after forming does not change, using the detailed parameters of the truss core panel shown in Table 1, the ratio of the average plate thickness before and after forming can be calculated as follows:

$$\frac{t_{after}}{t_{before}} = \frac{A_{before}}{A_{after}} = 0.45, \tag{9}$$

The truss core panel shown in Table 1 has a height of 33 mm. According to Equation (9), the ratio of the average plate thickness of the formed truss core panel to its thickness before forming is 0.45. This implies that after forming, the plate thickness will be too large. Forming is therefore not feasible.

### 3.4. Proposal for Practical Production

In the previous section, according to the processing procedure shown in Figure 10, the actual assembled truss core panel was manually processed, and its processing performance was examined.

For the future mass production of actual assembled truss core panels using the proposed method, we propose a processing method using a progressive die.

As shown in Figure 19, the proposed processing procedure includes five steps.

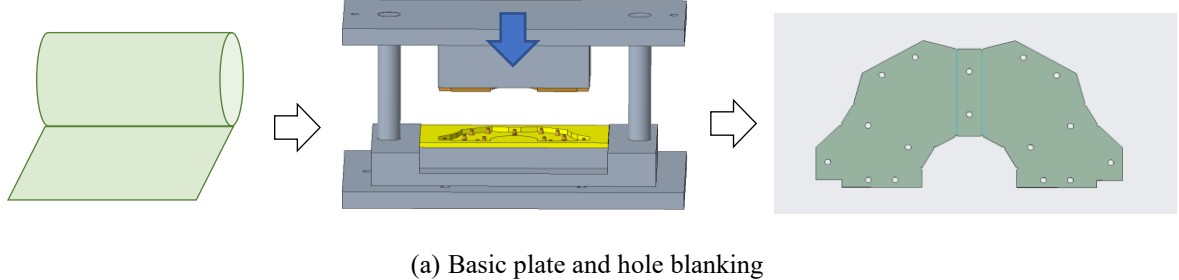

(a) Basic plate and hole blanking

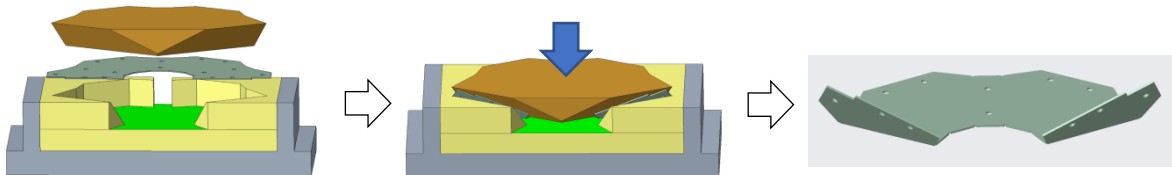

(b) Bending on both sides

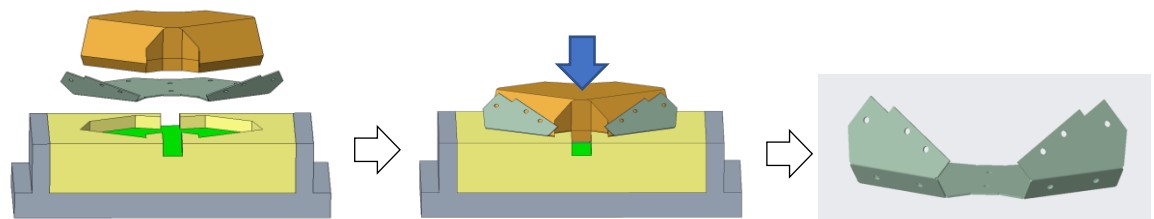

(c) Bending of the rectangular sides on both sides

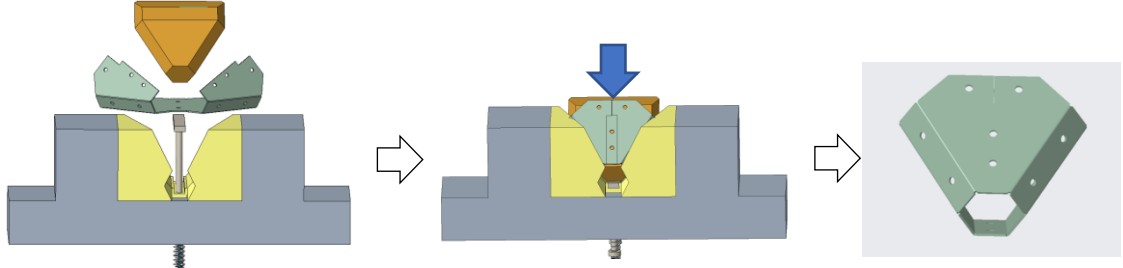

(d) Bending of the central rectangular side

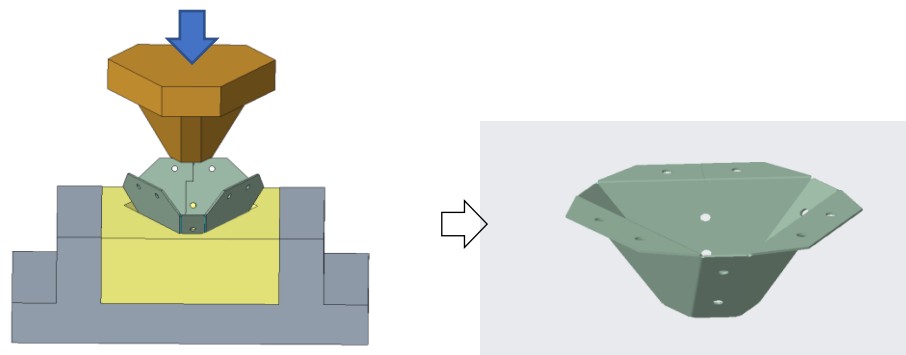

(e) Bending of the joint margin at the bottom

**Figure 19.** Processing plan using a progressive die for future mass production.

Step 1: Cut the truss core basic part, as shown in Figure 9, from the coiled thin strip steel with a punching die (Figure 19a).

Step 2: Bend both sides of the truss core basic flat plate part using a bending die to obtain a primary bent part (Figure 19b).

Step 3: Using a bending die, bend the part corresponding to the oblique rectangular side surface of the truss core from the primary bending part, which yields the secondary bent part (Figure 19c).

Step 4: Using a bending die, bend the central rectangular side portion of the truss core from the secondary bent part to obtain the tertiary bent part (Figure 19d).

Step 5: Continue using the bending die to bend the connecting wing sections from the tertiary bent part to the base of each lateral trapezoid to obtain a truss core component (Figure 19d).

Additionally, the processes of gluing and riveting can be combined to build a processing line for the entire assembled truss core panel structure.

The factors that determine the economic feasibility for commercializing the proposed assembled truss core panel consist of material, processing method, cost, and market. The material is a normal metal material and can be purchased at a relatively cheap price. Using a production line similar to the one shown in Figure 19, the processing method would be relatively simple and stable. The processing costs can be kept relatively low because the panels can be processed using simple methods such as punching and bending. The market potential includes applications such as the manufacture of the floors of electric vehicles, etc., and the product can be widely deployed. Taking into account these factors, the economic feasibility of the assembled truss core panel proposed in this study could be confirmed.

## 4. Conclusions

In this study, in order to solve the processing problem associated with the truss core panel when considered as a general-purpose lightweight panel, a new processing method for the assembled truss core panel was proposed. After conducting processing experiments and three-point bending tests, the following conclusions were drawn.

(1) An important design equation was derived for the practical use of the processing method for the assembled truss core panel. The design parameters a1, H, L, and those of the basic flat part can be calculated directly from the shape parameters a, b, and h of the truss core. The derived design equation was verified using experimental results.

(2) The processing experiment of an assembled truss core panel was carried out. A truss core part was obtained using simple punching and bending methods. Furthermore, it was joined using glue and rivets, and an assembled truss core panel was processed as expected. The validity of the proposed processing method was also confirmed.

(3) A three-point bending test was performed on the assembled truss core panel obtained in the processing experiment, and load deflection was measured. Assembled truss core panels joined by glue and rivets have a relatively high initial bending stiffness in the early elastic deformation and a relatively long-lasting bending deformation after the initial failure. After thorough testing, we concluded that the assembled truss core panel can be used as a general-strength panel structure.

(4) For comparison, FEM analysis was performed on the assembled truss core panel and honeycomb panel under the same conditions. The initial bending stiffness of the assembled truss core panel was found to be 10.60% higher than that of the honeycomb panel.

(5) The construction of a processing line using progressive dies was proposed in order to improve the efficiency of processing assembled truss core panels. The possibility of its practical application to mass production was thus demonstrated.

The assembled truss core panel proposed in this study can be designed with simple geometric parameters, it can be processed using a simple method, and it has a relatively high bending rigidity compared with that of conventional lightweight panel structures. Therefore, it can be used as a structural member, such as the floor of an electric vehicle or the bed of a truck. In addition, future practical research is expected to expand into industries such as construction and shipping.

In our future work, we will conduct practical research on a production line for the proposed assembled truss core panel. In addition, we plan to investigate ways of connecting the assembled truss core panel to frameworks in developing its application as a component of the floor structure of electric vehicles.

**Author Contributions:** Writing—original draft preparation, Z.T.; writing—review and editing, J.G. and X.Z.; data curation, C.K.; investigation, A.B.F. and W.Z.; software, C.K.; conceptualization, W.Z. and X.Z.; methodology, J.G. and Z.T.; validation, A.B.F. All authors have read and agreed to the published version of the manuscript.

**Funding:** This research received no external funding.

**Data Availability Statement:** Data available in a publicly accessible repository.

**Conflicts of Interest:** The authors declare no conflict of interest.

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
