# Peer review of "The Development of an Assembled Truss Core Lightweight Panel and Its Method of Manufacture"

_jmmp, doi:10.3390/jmmp7010029_

Round 1

Reviewer 1 Report

1. In this study, to solve the processing problem associated with the truss core panel when considered as a general-purpose lightweight panel, a new processing method for the assembled truss core panel was proposed. After conducting processing experiments and three-point bending tests. Its application as lightweight panels has been confirmed. To compare with the conventional honeycomb panel, FEM (Finite Element Method) analysis was performed on the assembled truss core panel and honeycomb panel under the same conditions. The bending stiffness of the assembled truss core panel was found to be 10.60% higher than that of the honeycomb panel. Overall, the paper structure and experiments are complete. The picture is clear and readable.

Reviewer 2 Report

General Comment

The manuscript proposes a new processing method to fabricate lightweight truss core panels as an alternative of the common press-forming method. The new processing method consists in cutting flat geometries from a plate, then the truss cores are obtained by bending the flat parts and by using glue and rivets for joining. The truss core panel is then obtained by joining the truss core and plates, again with glue and rivets.

The geometrical design and the assembly of the proposed truss cross panel are both presented and illustrated. Then, two truss core panels using aluminium were assembled according to the proposed methodology, and tested under three-point bending test to assess their mechanical behavior. The results are presented and discussed. In particular, the influence of using glue in addition to rivets is demonstrated due to the benefits of adhesion which ensure the spreading of the load. In addition, the comparison of the performance between the proposed assembled truss core panel and a honeycomb panel is studied by using finite element models. It is demonstrated that the proposed assembled truss core panel show a higher bending stiffness. Finally, a forming die proposal for the production of the proposed assembled truss core panel is described. In the end, the authors conclude about the viability of the proposed assembled truss core panel to be used as a general-strength panel structure.

The topic of the manuscript is interesting and important, since the processing of truss core panels still show some problems related with some mechanical weaknesses of the final product. The solution proposed in this study could be useful for future studies, for designers, and also for lightweight truss core panel suppliers.

I made some comments and suggestions to improve the article. The authors should take them into account and revise their article.

Specific Comment 1

The article requires a revision to correct some formatting issues. Some examples:

- A section should not start with a figure, and all figures should be cited in the text before they appear;

- the format of symbols should be the same in the text and in the figures (for instance: in italic);

- …

Specific Comment 2

Section 2.3, page 5

In the second paragraph, I think it should be “regular hexagonal top face” instead of “equilateral triangular top face”. See Fig. 6c. Please check.

Specific Comment 3

Section 2.4

Please present the properties of the used glue and discuss the type of glue to be used in order to ensure the required good adhesion during the service life of the panels.

Specific Comment 4

Section 3.1

In the text and in the figures, please avoid to use the words “pressure” and “compression”. For instance: “pressure displacement”, “compressive deformation”, “compressive load”, … Also, it should be “initial bending stiffness” instead of “bending stiffness”.

Specific Comment 5

Section 3.2

Please provide more information about the finite element models. For instance: used software, used finite elements, mesh density, … How the glue and rivets were modelled?

Specific Comment 6

Section 3.2

The numerical model for the assembled truss core panel should be calibrated and validated using your previous experimental results.

Specific Comment 7

Section 3.2

For the comparison between the performance of the assembled truss core panel and the honeycomb panel models to be considered valid, some property should be identical for the two models (for instance, the quantity or weight of the used materials). However, this is not discussed in the manuscript and doubts can arise to the readers about the validity between the performed comparisons.

Specific Comment 8

Section 3.2

From my understanding, only an elastic analysis was performed with the finite element models. This is very limiting! The analysis should be more complete in order to present for each model the load – displacement curve, at least until de maximum load is reached. This would require a non-linear finite element analysis. These additional result would greatly improve the quality of Section 3.2.

Specific Comment 9

Section 3.2, page 13

In the paragraph after Figure 16, please explain “Under the same analytical conditions”. You are performing a numerical analysis and not an analytical one.

Specific Comment 10

Section 3.3

I didn’t understand well your conclusion in the end of this section. Please improve this part and explain better.

Specific Comment 11

Section 3.4

Would it be possible to complement this section, or add a new one, with a simplified economic feasibility study to support your conclusion about the viability of the proposed assembled truss core panel? This would be important!

Specific Comment 12

Section 4

Please add a paragraph with the needs for future studies in order to complement your results and help to support your conclusion about the viability of the proposed assembled truss core panel to be used in real life service.

Reviewer 3 Report

The reviewer thanks the authors for their contribution which has the goal to solve a particular problem in the field of structural engineering. In general, the experimental and numerical work is appreciated, and the findings are useful information. Nevertheless, the publication in the “Journal of Manufacturing and Materials Processing, MDPI” is not recommended unless the following suggestions are taken into account:

1) Please, polish the abstract. Please, add sentences to explain the meaning, the main points, the improvement and the promising application of the study. The logic should also be improved.

2) Introduction. The current state of knowledge relating to the manuscript topic has not been covered and clearly presented, and the authors’ contributions are not emphasized. In this regard, the authors should make their effort to address these issues, by adding additional comments on the state of the art and the proposed aspects.

3) Introduction. In the literature, the use of aluminum alloys instead of metallic materials has recently been studied and verified for space frames and trusses. Please, also refer to the following references:

-  https://doi.org/10.1142/S021945541850092X

https://doi.org/10.1016/j.tws.2017.02.008

4) Objectives and findings should be presented more clearly (e.g., using the following division of the sections: Introduction, Proposed Technology, Case of Study: Laboratory Tests, Case of Study: Finite Element Numerical Modeling, Comparison between Laboratory and Numerical Tests, Results and Discussion, and Conclusions). The current sections appear not well organized and divided. Furthermore, comments should be added in regard to the practical value of this work, and how the industry can profit from this manuscript.

5) Laboratory Tests. Please, report the test layout of experiments with locations of loading points and devices used. Please, specify.

6) Laboratory Tests. Please, specify the corresponding characteristics of the measurements, e.g., frequency (or the period) of the recording data etc. Moreover, range (mm), sensitivity (pm/mm), resolution (mm) and accuracy (mm) of the recording data by all the used equipment and devices should be underlined as well as with their technical characteristics.

7) Finite Element Numerical Modeling. The finite element (FE) analyses must be better explained with, particularly, details of the models used. It is not clear how the models of the panels are composed (beam elements, plate and shell etc., with the corresponding amounts and mesh sizes) and how the external loading were applied. Which are the geometric characteristics of the panels ? Please, revise these parts and provide more information about the models.

8) Finite Element Numerical Modeling. Please, insert a table which lists the types of FE used, with the corresponding amounts within the models and mesh sizes.

9) The deformations obtained from FE modeling and devices should be listed within a table with the corresponding comparison errors. Moreover, which software has been used for the FE modeling ? Please, cite the software within the references.

10) The further work related to this article is important and should be emphasized in the conclusions.

Round 2

Reviewer 2 Report

I received and read the revised version of the manuscript with title “Development of assembled truss core lightweight panel and its manufacturing method”. I´m generally satisfied with the authors’ replies to my comments and I consider that most of the recommendations were considered to improve the manuscript.

Reviewer 3 Report

The authors carried out the required revisions.